# The Winner Takes It All: Auxin—The Main Player during Plant Embryogenesis

**DOI:** 10.3390/cells9030606

**Published:** 2020-03-03

**Authors:** Konrad Winnicki

**Affiliations:** Department of Cytophysiology, Faculty of Biology and Environmental Protection, University of Lodz, 90-236 Lódź, Poland; konrad.winnicki@biol.uni.lodz.pl

**Keywords:** MAPK, embryogenesis, cellular patterning, microtubules, actin filaments, auxin, calcium ions

## Abstract

In plants, the first asymmetrical division of a zygote leads to the formation of two cells with different developmental fates. The establishment of various patterns relies on spatial and temporal gene expression, however the precise mechanism responsible for embryonic patterning still needs elucidation. Auxin seems to be the main player which regulates embryo development and controls expression of various genes in a dose-dependent manner. Thus, local auxin maxima and minima which are provided by polar auxin transport underlie cell fate specification. Diverse auxin concentrations in various regions of an embryo would easily explain distinct cell identities, however the question about the mechanism of cellular patterning in cells exposed to similar auxin concentrations still remains open. Thus, specification of cell fate might result not only from the cell position within an embryo but also from events occurring before and during mitosis. This review presents the impact of auxin on the orientation of the cell division plane and discusses the mechanism of auxin-dependent cytoskeleton alignment. Furthermore, close attention is paid to auxin-induced calcium fluxes, which regulate the activity of MAPKs during postembryonic development and which possibly might also underlie cellular patterning during embryogenesis.

## 1. Introduction

Embryogenesis is driven by sophisticated and orchestrated processes which are kept under the control of various molecular pathways. Phenotypic analyses of mutants and studies with the use of protein inhibitors provide clear evidence that three molecular pathways are indispensable for a correct embryo development, and these include polar auxin transport, mitogen-activated protein kinases and DNA methylation. Dysfunction of these molecular pathways leads to embryogenesis disorders and may be lethal for the developing embryo [1,2,3]. In angiosperms after fertilization, a zygote elongates in micropylar-chalzal axis and divides asymmetrically. In this way the extraordinary process begins which leads to development of the functional embryo [4,5,6]. Since auxin was found to be responsible for cell extension according to the direction of auxin flow [7], it seems that the polar auxin transport may determine the direction of the zygote elongation. Although the disruption of the polar auxin transport manifests itself mainly during the apical-basal axis formation, some disorders of embryogenesis are also noticeable at earlier stages [1,8,9]. The process of zygote elongation is crucial for the first asymmetrical division, as a result of which a small apical and a large basal cells are created, both already with different gene expression [10,11,12]. Detailed research on the plant embryogenesis indicates that the asymmetrical division of the zygote relies both on microtubules, which control the zygote elongation, and on actin filaments, which regulate polar migration of nuclei [13,14].

After early cell divisions, the embryo starts to develop specific tissues. Different tissues were found to have unique profiles of activated genes [10,15,16,17], however the precise mechanism responsible for the diverse gene expression needs elucidation and the question when and how cellular patterning is established still remains open. The general plane of embryogenesis and the accuracy of this process incline to state that not only molecular pathways mentioned above but also mutual relationships between them and other events should be taken into consideration. Since auxin is a plant phytohormone controlling the cell polarity, the orientation of the cell division and expression of auxin-dependent genes, one could ask how auxin regulates cytoskeleton alignment and establishment of various gene expression during embryogenesis. This review presents cytophysiological responses to auxin which affect cytoskeleton alignment, and show that gene expression may contribute to cellular patterning.

## 2. Role of Cytoskeleton in Embryo Development

With the first division of a zygote, embryogenesis follows the patterns which allow it to develop a final profile of an embryo. Thus, mechanisms which regulate cell shape and interactions between them must be taken into consideration. Replication and cell division are energy consuming processes, therefore it is not surprising that cells tend to reduce energy expenditure during their lifespan. The shortest wall rule assumes that cell wall after mitosis is created according to a surface which needs minimum energy [18,19,20,21].

Throughout anisotropic growth cells display transverse pattern of cortical microtubules. This alignment was found in developing trichomes [22,23] and in a root elongation zone [24]. Thus, rectangular cells usually divide transversely, however, there are exceptions to this pattern during morphogenesis, and response to mechanical stress is a good example when the shortest wall rule is broken [25,26,27]. Mechanical stress regulates microtubule orientation at subcellular and tissue levels and they were found to align along the direction of maximal stress in epidermal cells of *Arabidopsis thaliana* [27]. Interestingly, orientation of cortical microtubules was found to be parallel to localization of PIN-FORMED (PIN) proteins, one of the plasma membrane auxin transporters. Mechanical stress induces both circumferential localization of microtubules around the site of cell ablation and translocation of PIN proteins to membranes which are away from the place of damage, however parallel alignment of microtubules and PINs was maintained [28,29]. During morphogenesis, mechanical stress within tissues may be induced when neighboring cells grow faster and PINs seem to be recruited to the membrane with the highest tensile stress [30]. Explanation of this behavior may come from the mechanism of PINs cycling between cell membranes which utilize endocytosis and exocytosis. It was found that high tension of plasma membrane inhibited endocytosis, however exocytosis might reduce the emerging tension. Therefore, PINs density was found to increase when plasma membrane tension enlarges [31]. Thus, it is very plausible that tension stress regulates microtubule alignment during embryogenesis and might be responsible for breaking the rule of division along the shortest cell wall. Furthermore, parallel alignment of microtubules and PIN proteins indicates that auxin must cooperate in this process.

Actin filaments (F-actin), another cytoskeleton component, were found to play a role in the directional cell growth as well. Two fractions of F-actin bundles exist inside cells, one which decorates plasma membrane (cortical F-actin) and the other which polymerizes in cytoplasm [32]. Longitudinal bundling of cytoplasmic F-actin takes part in trichomes development, root hair growth [22,23,33], pollen tube expansion [32,34] and anisotropy growth of hypocotyl cells [24]. Depending on localization, F-actin bundles display different impact on cellular molecules, vesicle transport and finally cell polarization. Cortical F-actin bundles seem to act as a physical barrier for vesicle docking and transport of molecules through plasma membrane. On the other hand, directional growth of cytoplasmic F-actin bundles was found to be responsible for polarized trafficking. Thus, cortical and cytoplasmic F-actin bundles co-regulate directional transport of molecules, creating a physical barrier and triggering polar localization of plasma membrane transporters [35,36,37,38,39,40]. Interestingly, cortical microtubules and actin filaments were found to coalign [32,41], which may indicate their mutual dependence [24,42]. It seems to be very plausible that the alignment of cortical microtubules and F-actin bundles by plasma membrane reduces transport at the site of their polymerization and together with cytoplasmic F-actin, they support directional transfer of molecules and finally anisotropic growth. However, the question of whether the opposite alignment of cytoplasmic F-actin and cortical microtubules depends on each other still remains open.

It has been previously shown that microtubule alignment is strictly connected with the division plane between 2- and 16-cell stages of plant embryogenesis [43,44]. Thus, it seems that elements of cytoskeleton with their impact on mechanical properties of cells and signaling pathways which control cytoskeleton alignment must play a prominent role in the determination of the division plane during embryogenesis. Studies on embryo development indicate that similarly to post-embrionic growth, it relies on mitoses which are beyond the shortest wall rule, and the first two divisions of the small apical cell in *A. thaliana* embryos take place in the longitudinal plane [21]. It is likely that tensile stress impacting the alignment of microtubules and actin filaments regulates the directional transport, the cell polarization and finally the direction of future division. This mechanism may be conservative and takes part both in embryo development and in post-embryonic growth. However, it seems that the response to mechanical stress solely may not be the only mechanism responsible for regulation of cell division plane and cellular patterning.

Microtubule interactions such as zippering, catastrophe or crossover, which underlie changes in cytoskeleton alignment, seem to impact the correct embryo development as well. The first term describes the state when two microtubules meet at an angle smaller than 40° and continue to growth next to each other, the second when microtubules shrink after meeting at an angle larger than 40° and the last when two microtubules slip over each other. Computational analyses indicate that the probability of transverse and longitudinal alignment is nearly equal, however the situation dramatically changes when edge-catastrophe and enhanced stability of microtubules at new division site are taken into consideration [43,44]. The edge-catastrophe, i.e., shrinking of the microtubules which reached cell edge and high probability of their collision make molecular apparatus regulating microtubule dynamics indispensable during growth along the shortest cell wall. Due to the limited amount of free tubulin and the dynamic instability [45], short microtubules with short lifespan would be more favorable when edge-catastrophe is diminished. Thus, transverse alignment seems to be the default orientation of microtubules in cells revealing anisotropic growth.

## 3. Cytoskeleton Elements are Regulated by Auxin

Auxin was postulated to be the main player which regulates microtubule growth, however some opposing results have been demonstrated so far. Although exogenous auxin triggers the transverse position of cortical microtubules in hypocotyl cells [46,47], contradictory studies indicate that an increase in auxin concentration induces inhibition of cell elongation [48] which is accompanied by reorientation of cortical microtubules from transverse to longitudinal alignment. This behavior was observed in the transition zone of root and in the elongation zone of hypocotyl. Inhibition of cell elongation in response to auxin relies on auxin binding protein 1 (ABP1) [49] and it was found to be non-transcriptional-dependent [46,49,50]. On the other hand, aberrant horizontal divisions of an apical cell and embryogenesis disorders in *mp* (auxin response factor) and *bdl* (Aux-IAA repressor) mutants [51,52] may lead to the conclusion that regulation of microtubule alignment is transcriptional-dependent. However, it must be taken into consideration that embryos with those mutant proteins display impaired polar auxin transport which prevents establishment of the correct auxin maxima. Thus, it appears that microtubule alignment depends indirectly on the expression of auxin-inducible genes which are responsible for final auxin localization, and the direct regulation of microtubule ordering seems not to depend on transcription.

A high auxin level induces cortical microtubule disorganization during anisotropic expansion and supports the shift into isotropic cell growth [18,53,54]. Microtubule alignment along the longest cell wall after exogenous auxin application might be extrapolated to embryogenesis and would explain orientation of the division plane in embryonic cells. In *A. thaliana* after the first division of a zygote, auxin is transported toward the apical cell due to polar localization of PIN7 proteins [55]. The apical cell elongates slightly in the direction of auxin flow [56,57,58,59], however progressive auxin accumulation [60] would break default transverse alignment of microtubules and promote longitudinal division in this way. Interestingly, anisotropic growth of the apical and basal cells seems to be necessary for correct embryogenesis. A mutation in genes encoding polygalacturonases, enzymes degrading pectins, reduces the length of those cells leading to slower embryogenesis [59]. Furthermore, the regulation of a division plane orientation through the creation of auxin maxima may be supported indirectly by the studies which indicate longitudinal mitoses of the apical cell regenerated from a suspensor [61,62].

Auxin regulates not only microtubule alignment but also the growth of cytoplasmic F-actin filaments. Although there are no studies showing the impact of auxin level on direction of F-actin growth, auxin was found to play a role in actin filament rearrangements or actin-dependent transport. It was suggested that auxin regulated the organization of fine F-actin bundles, providing an efficient transport of signaling proteins toward the cell pole, and reduction of polar auxin transport was found to initiate actin filament bundling which may detain polar distribution of auxin transporters [34,63,64,65,66]. Thus, dissociation of actin bundles might be necessary for redistribution of PIN proteins to the plasma membrane [63,64]. Furthermore, auxin induces the production of phosphatidic acid which may regulate the organization of actin filaments and the vesicle trafficking, therefore it might have impact on the final PIN localization [67].

Since auxin regulates alignment of cytoskeleton elements and exogenous auxin was found to induce the microtubule reorientation from transverse to longitudinal during postembryonic development [46,49,50], analogous role of auxin is possible during embryogenesis. Thus, one could ask what are the players which stand behind the shift from transverse to longitudinal plane of an embryonic cell division?

## 4. The Players Regulating Auxin-Dependent Microtubule Alignment During Embryogenesis

Computational analyses show that the auxin-mediated microtubule stability at a new division site and the CLASP-mediated reduction of edge-catastrophe are the key factors regulating the division plane orientation during early stages of embryogenesis [43,44]. Nevertheless, the authors of these papers propose that alignment of cortical microtubules may also depend on the direction of mechanical stress at further stages of embryogenesis and they do not exclude both mechanisms acting synergistically. Enhanced microtubule stability at a new division site may explain their longitudinal alignment and the vertical division of the apical cell in *A. thaliana*, however it does not seem to be the general rule since the apical cell in some species divides horizontally [68,69,70]. Transverse microtubule bundles relay on the CLASP proteins which stabilizes plus-end of severed microtubules [71] and protects them from a catastrophe when they reach cell cortex [46,50]. However, transverse microtubule alignment is maintained in clasp-1 mutants during postembryonic growth, and only their abundance was found to be reduced [72]. Furthermore, a dysfunction of CLASP protein does not affect vertical divisions of the apical cell at the beginning of embryogenesis [43,44]. Thus, it seems reasonable to ask how auxin regulates microtubule alignment and the orientation of the division plane in embryonic cells.

It is possible that other proteins regulating microtubule behavior are responsible for cytoskeleton rearrangements during embryo development and I concentrate on two proteins whose specific functions during embryogenesis still need elucidation. Katanin, the microtubule-severing protein, was found to be indispensable for the transverse microtubule alignment during postembryonic development [73,74] and disorders of embryogenesis in katanin mutants were previously shown [75]. Although auxin activates the katanin activity via the ROP6-RIC1 pathway [54,73,76,77], a high auxin level induces calcium release from intracellular reservoirs and the external space [78] which at an elevated concentration may inhibit the katanin activity without changing its ability to bind substrates [79]. Therefore, it is possible that due to high auxin level in the apical cell, katanin activity is reduced and the transverse alignment of microtubules is not supported. Calcium ions were also found to accelerate the activity of MDP25, one of the microtubule destabilizing proteins. Consequently, a high auxin level both reduces the severing activity of katanin and intensifies the MDP25 destabilizing action, leading to a higher frequency of an encounter-based catastrophe of transversely growing microtubules, and therefore making the longitudinal ordering more favorable (Figure 1A). Furthermore, the depolimerization of short microtubules increases the amount of free tubulin, which may be utilized by those microtubules growing longitudinally and having a smaller chance for collision. An oblique or longitudinal array of microtubules in the cells overexpressing MDP25 may support the role of this protein in the shift from transverse to longitudinal alignment [80,81]. Auxin was also found to increase the level of phosphatidic acid which mediates through MAP65 (one of microtubule bundling protein) attachment of microtubules to plasma membrane [82,83], and therefore it could promote higher stability of longitudinally growing microtubules.

Although some studies indicate that PIN localization and microtubule alignment are independent of auxin in specific situations [28,29], a transcriptional response to auxin and polar auxin transport were found to be necessary both for overriding the shortest wall rule and for asymmetric division, which ensure the correct embryo development [21]. A mechanical stress and an auxin-dependent response were proposed as the two mechanisms responsible for changes of cell shape in a shoot apical meristem [84]. It seems that auxin plays a crucial role in correct embryogenesis and mechanical stress may act as a complementary mechanism modulating the orientation of the division plane or might be switched on when necessary.

If the mechanism employing katanin and MDP25 protein is responsible for longitudinal alignment of microtubules during embryogenesis, one could ask how microtubule ordering is changed to drive a horizontal plane of the cell divisions. Since high intracellular level of calcium ions is temporary [85], it is possible that after two longitudinal divisions of the apical cell, the level of free calcium ions declines or the activation of calcium ion channels is not intensive enough, so that the severing function of katanin is maintained and the destabilizing activity of MDP25 is silenced. This may increase the stability of short microtubules and support the transverse microtubule alignment (Figure 1B).

## 5. Calcium Ions at the Service of Auxin

Auxin induces a dose-dependent depolarization of plasma membrane which results in calcium influx. It was also found to activate a plasma membrane H+-ATPase which pumps H+ ions to apoplast, and thereby provides a low pH in the extracellular space. Since apoplast acidification increases the membrane potential, the activity of the H+-ATPase may be considered as one of the mechanisms which regulate the extent of calcium influx. Interestingly, although the membrane depolarization continued, calcium concentration was found to return to the baseline level [86,87,88,89]. Elevation of calcium ion levels in cytoplasm results in their subsequent release from vacuole [90,91]. Auxin seems to participate in amplification of calcium signaling, since it activates a phospholipase C (PLC) [82] whose optimal activity depends on calcium ions [92,93]. The PLC hydrolizes phosphatidylinositol 4,5-bisphosphate (PIP2) to inositol-1,4,5-triphosphate (IP3) which is a messenger molecule. IP3 by binding to receptors in the endoplasmic reticulum (ER) induces calcium efflux from the lumen [67,94,95]. Furthermore, starting from one cell embryo, small vacuoles are present in cytoplasm [96,97] and they may also be a source of calcium ions. The tonoplast is decorated with voltage-dependent slow vacuolar (SV) channels and ligand-gated channels. The former may be activated by calcium ions and auxin was found to enhance these channels activity. The latter, similarly to ER, are activated by IP3 [98]. Interestingly, calmodulin, an adaptor of calcium signals, was found to modulate SV channels activity [99]. Calcium signaling during plant embryogenesis is not studied in detail, however it has been shown so far that calcium ions [100] and calmodulin [101,102] are indispensable during somatic embryogenesis. Thus, considering calcium oscillation during postembryonic development, it is worth asking whether two steps of calcium ions accumulation could take place also during embryogenesis. A hypothetical model of calcium influx which could impact cytoskeleton alignment during embryogenesis is presented in Figure 2. First, the calcium level elevates due to fluxes through plasma membrane and after it reaches an adequate level inside a cell, calcium efflux from vacuoles and ER is activated. Thus, it is possible that due to a progressive accumulation of auxin in a apical cell and in daughter cells after the first division, the concentration of calcium ions and the time of their action are sufficient to impact microtubule regulating proteins and to induce the longitudinal alignment of microtubules (Figure 1A and Figure 2A).

This model might also explain the shift to transverse alignment at further stages of embryogenesis. Following two longitudinal mitosis, the volume of an apical cell becomes nearly 4-fold smaller [21]. Theoretically, the surface of the basal membrane, which ensures directional auxin transport, is also 4-fold smaller, however, the surface of vertical membranes of daughter cells is only halved. Thus, it seems possible that the level of calcium ions is reached much faster in daughter cells after the second division of the apical cell and their high concentration in cytoplasm triggers the inhibition of plasma membrane H+-ATPase much earlier, leading to the cytosol acidification and apoplast alkalization, similarly as it happens during postembryonic development [103,104,105]. Under low intracellular pH, both voltage-dependent channels in a vacuole are inhibited [106] and auxin appears in the protonated form (IAAH) which, due to lipophilic properties, may cross lipid bilayer [107]. It is possible that the cytosol acidification decreases the auxin concentration and reduces the activation of calcium channels in ER or vacuole, thereby prevents further accumulation of calcium ions and their long-term action (Figure 2B). Thus, under low calcium level severing katanin activity is maintained in contrast to MDP25 activity. This provides the transverse microtubule alignment, which could result in the horizontal direction of the third division (Figure 1B).

## 6. Asymmetric Division Depends on Auxin

The asymmetric division which generates daughter cells with unequal volumes may have an impact on the cell fate during embryogenesis. At a 16-cell embryo stage, the asymmetrical mitosis occurs in an outer and inner cell of the lower tier. The former divides periclinally, giving a larger apical and smaller basal cell, while the latter divides anticlinally, generating a larger outer ground tissue precursor and a smaller inner vascular precursor [21]. The orientation of cell division, especially at further stages of embryogenesis may depend on auxin gradient and tensions generated between cells [44]. As far as both mechanisms explain the direction of cell division, could they elucidate the choice between symmetrical and asymmetrical divisions?

Although alignment of cortical microtubules and the formation of preprophase band were found to dictate the direction of cell division, actin filaments seem to regulate the polar migration of nuclei which heralds upcoming unequal mitosis [13,108]. Furthermore, in animal and yeast cells, actin filaments were found to position the mitotic spindle, and thus to have an impact on the symmetry of cell division [14]. Unequal mitosis during stomata or root development depends on auxin and PIN proteins, and inhibitors of a polar auxin transport were found to restrain the polar migration of nuclei in plant cells [109,110,111]. It seems that the choice between symmetrical or asymmetrical mitosis depends on actin filaments whose alignment is regulated by auxin. Although the direction and the symmetry of cell division depend on cytoskeleton alignment and auxin transport, this does not explain directly the mechanism in which the spatiotemporal expression of specific genes is established during embryo development.

## 7. New Mechanism Regulating Intracellular Asymmetry and Cellular Patterning

Cellular patterning starts very early during embryogenesis and already after the first division of a zygote different genes are expressed in apical and basal cells. Many genes are known to be differently expressed in various cells and tissues and their impact on the cellular patterning is well established [4,6,10,112]. Following two longitudinal divisions of an apical cell, the same genes were found to be activated in embryo proper, however, after transverse divisions, when an embryo reaches 8-cell stage, upper and bottom tiers display different gene expression which is controlled by auxin threshold [60,113,114]. Thus, auxin transporters, whose expression is induced by high auxin level [115,116], regulate polar auxin transport and thereby generate auxin maxima in different tissues. A family of auxin transporters is a good example of genes differently expressed in various cells. LAX1, one of the auxin influx carriers, is restricted to embryo proper and once an embryo reaches 32-cell stage, it shows higher expression in the upper tier. LAX2 was not detected at globular stage, however following the 32-cell stage, this protein is restricted to provascular cells and hypophysis. PIN proteins, auxin efflux carriers, also display cell-specific localization. For example, PIN7 is restricted to suspensor, PIN4 to hypophysis and PIN1 was generally found expressed in embryo proper. However, PIN1 displays no polar localization before 16-cell stage and its polarity is established during the next steps of embryogenesis. Interestingly, PIN1 is mainly localized in a basal membrane of provascular cells and in an apical membrane in outermost cells [1,5,8,9,10,11,55]. Another example of genes differently expressed in various cells is the WOX gene family which encodes transcription factors that regulate plant development. WOX9 and WOX2 were found to be expressed in an apical and central domain but WOX8 in hypophysis and suspensor only [6,16,60].

Expression of various genes in particular regions of a developing embryo raises a question about the timing and the way in which these cellular patterns are established. One could ask whether they are set up after the division as a result of the cell position and hence different auxin level or maybe before mitosis as a result of intracellular auxin gradients.

The existence of embryo regions with different auxin levels may lead to the conclusion that the cell position within an embryo plays a major role during cellular patterning. Furthermore, inversion of root meristem to shoot meristem or changing the shoot meristem patterns to root meristem which depend on auxin level [117,118,119,120] may support this idea. Although cell position and the resulting exposure to different auxin concentrations explain well cell fate at the level of organs, it does not explicate the development of cells with different identities, which are located in a region displaying homogenous auxin level.

It was previously described that MAPKs were indispensable during embryogenesis and that mutations in genes of MAPK pathway lead to reduced elongation of zygote and symmetrical cell division [4]. MAPKs control expression of different genes through interaction with transcription factors or regulate other cellular processes by targeting cytoplasmic proteins, e.g., in rice MAPKs were found to interact with Aux/Lax1 [121,122,123]. Studies on *Vicia faba* embryos revealed a new and so far not described localization of MAPKs. The phosphorylated form of these kinases was found in the vicinity of chromosomes and auxin seems to regulate the number of mitoses with MAPKs and the number of spots with activated kinases localized on the surface of chromosomes. Interestingly, the number of spots differs between groups of separating chromatids during anaphase and telophase [124]. This phenomenon raises the question about the mechanism which could regulate unequal distribution of MAPKs in the vicinity of chromosomes. During postembryonic development and stress response, phosphorylation of MAPKs is controlled by dual-specificity phosphatases (DUSPs). Calmodulin, which is a calcium sensor, was found to increase the activity of dual-specificity phosphatases (DUSPs) indicating a link between calcium ions oscillation and MAPKs phosphorylation [125,126]. Asymmetrically distributed calcium ions were well documented in highly elongated cells such as growing pollen tubes [127,128], and exogenous auxin was found to induce calcium ions release to cytoplasm [78]. However, intracellular gradients of calcium and auxin are difficult to follow during plant embryogenesis. Nevertheless, intracellular auxin gradient was detected directly in cells during proliferative period of *Chara vulgaris* [129] and this gradient might exist in embryonic cells as well.

It is possible that, due to auxin gradient, locally activated DUSPs dephosphorylate MAPKs in their conservative TXY motif, and thereby reduce the number of activated MAPKs in the vicinity of those chromatids which are exposed to a high calcium content. Calcium dependent dephosphorylation of MAPKs may also be supported by studies showing that the function of PP2C kinase, another potential MPAKs phosphatase, is regulated by calcium-dependent protein kinase (CDPK) [130]. However, some studies indicate a reduced activity of PP2C under a high concentration of auxin or calcium ions (Figure 3A,B) [131,132,133]. Another explanation of asymmetric MAPK distribution may come from regulation of chromatin structure or segregation of molecules in cytoplasm. Chromosomes exposed to calcium ions display compact structure [134], thus it is possible that the level of chromatin condensation might differ between groups of chromatids due to calcium gradients, and in this way calcium ions could facilitate or reduce DNA accessibility for MAPKs. Thus, auxin-induced calcium fluxes might regulate the number of phosphorylated MPAKs in the vicinity of chromosomes, as they might trigger the activation of phosphatases or they might control chromatin compaction which reduces accessibility of DNA for MAPKs. Interestingly, calcium ion gradient might induce an electrical field which would segregate molecules by electrophoresis [135,136] and this could be another mechanism regulating protein composition at both ends of a dividing cell. It is very plausible that MAPKs participate in cellular patterning before cell division is completed, and due to their unequal distribution in separating chromatids, they execute the asymmetry of intracellular gradients of auxin and calcium molecules. Mammalian transcription factors (TFs) were found to associate with mitotic chromosomes and their role in regulation of cell fate by controlling post-mitotic gene reactivation was previously postulated [137,138]. It seems that MAPKs may act in the same way as pioneer TFs and mark the genes which will be activated in daughter cells early during mitotic exit.

## 8. Conclusions

Cellular patterning, which starts with the first division of a zygote, seems to be executed not only by regulation of different gene expression but also through factors which control the direction and the symmetry of cell division. Auxin, which is multifunctional plant phytohormone, regulates post-embryonic development and response to stress stimuli. They were also found to play a considerable role in cellular patterning during embryogenesis. Although many genes and molecular pathways regulated by auxin are well known, precise mechanisms which underlie auxin-dependent control of embryogenesis still need elucidation. Auxin was found to regulate the release of calcium ions to cytoplasm and it is possible that their accumulation in cytoplasm takes place in two stages. Thus, the direction and the symmetry of cell division which is controlled by cytoskeleton alignment might depend not only on high level of calcium ions, but also on the duration of their action. Furthermore, chromosomal localization of MAPK might indicate a new function of these kinases during cellular patterning. Nevertheless, exactly which of many MAPKs identified in a plant genome play the prominent role in this process and how MAPKs in the vicinity of chromosome impact gene expression is yet to be determined.

## Figures and Tables

**Figure 1 cells-09-00606-f001:**
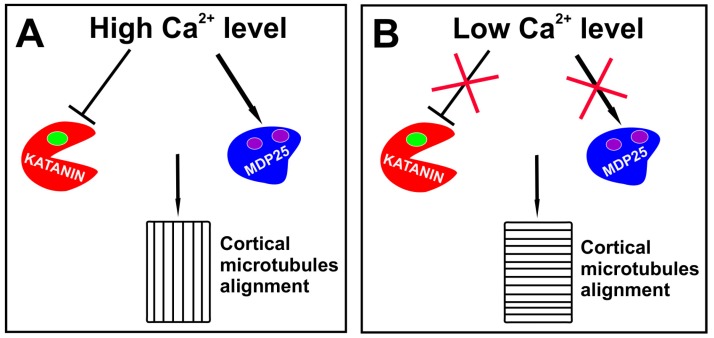
The role of calcium ions in the regulation of the direction of cell division during embryogenesis. (**A**) A high concentration of calcium ions negatively regulates functioning of katanin and activate destabilizing properties of MDP25, finally supporting longitudinal alignment of microtubules. (**B**) Low concentration of calcium ions has limited impact on katanin or MDP25. In this situation katanin activity is maintained and function of MDP25 is stopped, which results in transverse alignment of microtubules.

**Figure 2 cells-09-00606-f002:**
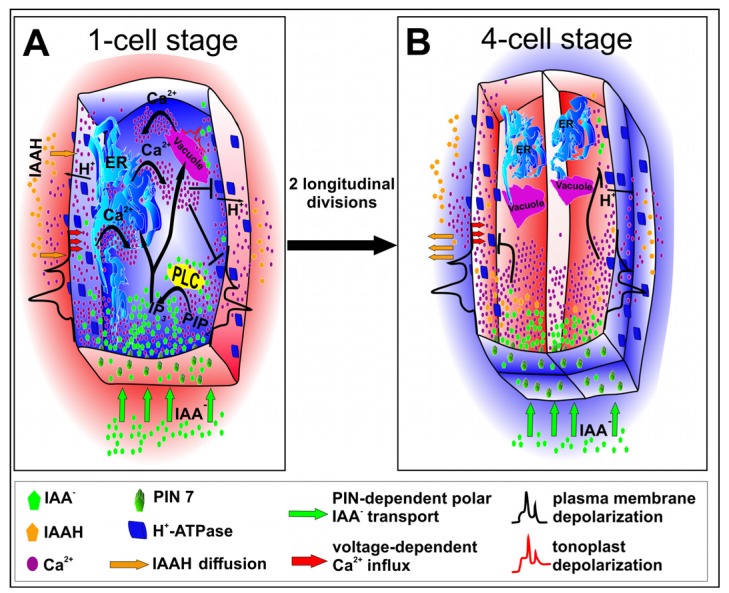
A hypothetical model of calcium ion release. (**A**) Transport of auxin ions to the apical cell is provided by polar localization of PIN7 proteins. The auxin influx induces plasma membrane depolarization, followed by an opening of calcium channels. Auxin and calcium ions regulate the activity of phospholipase C (PLC) which catalyzes conversion of PIP2 to IP3. IP3 induces opening ligand-gated calcium channels by binding to specific receptors in ER or vacuole. An elevated concentration of calcium ions activates also voltage-dependent channels in a vacuole. On the other hand, high concentration of calcium ions triggers inhibition of plasma membrane H+-ATPase which results in apoplast alkalization. High pH of an apoplast reduces auxin-induced plasma membrane depolarization and may act as self-attenuating mechanism of auxin impact. (**B**) After two longitudinal divisions the cell volume is 4-fold smaller but the surface of vertical membranes of daughter cells is only halved. All this results in faster achievement of high calcium concentration and following inhibition of plasma membrane H+-ATPase. Apoplast alkalization reduces auxin-induced depolarization of plasma membrane. In turn, cytosol acidification results in auxin protonation which allows for its diffusion from cytoplasm. Thus, both auxin concentration and calcium ion release from ER and vacuole are diminished.

**Figure 3 cells-09-00606-f003:**
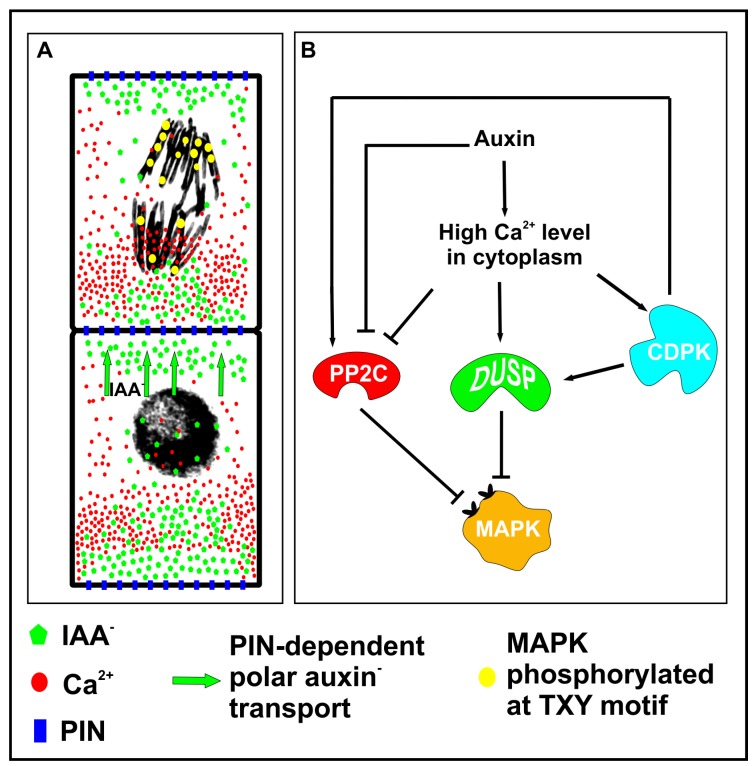
A hypothetical model of mitogen-activated protein kinase (MAPK)-dependent cellular patterning during embryogenesis. (**A**) The asymmetrical distribution of auxin, calcium ions and phosphorylated MAPKs in the vicinity of chromosomes. (**B**) High concentration of calcium ions at one end activates dual-specificity phosphatases (DUSPs) which dephosphorylate MAPKs and diminish the number of activated kinases in the vicinity of chromosomes. However, low calcium level at the opposite end might result in reduced activity of phosphatases, and therefore the number of chromatids with activated MAPKs might be greater at this end. Interestingly calcium ions activate CDPKs (calcium-dependent protein kinases) which may positively regulate PP2C phosphatase, however high calcium concentration was found to directly inhibit the activity of PP2C phosphatase.

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
