# Peer review of "The Winner Takes It All: Auxin—The Main Player during Plant Embryogenesis"

_cells, 2020, doi:10.3390/cells9030606_

Round 1

Reviewer 1 Report

line 65:

Would you like to describe the shortest wall rule?

Please explain the zippering, catastrophy and crossover of microtubules.

Would you like to explain edge-catastrophe and enhanced stability?

line 112:

Which other mechanisms do you propose?

line 151:

it is not clear. With the same orientation?

line 198:

In which way, the reduction of katanin activity would be responsible for the shift in the orientation of what?

In Fig. 1-B

It is difficult to understand the role of katatin. Is it activated at low Ca2+ concentration?

line 235:

it is not clear, revise the English

line 2011nd 202.

This idea is not clear.

line 392:

Check the English 

Author Response

Dear Reviewer,

I thank for the critical reading of the manuscript and your comments which made me think about highlighted issues. According to the Academic Editor some changes in the manuscript have been made. Please be also aware that in Figure 3 I changed depiction one of the symbol  (from PIN7 to PIN).

Furthermore, all changed and rewritten parts of the manuscript are highlighted. Some blocks of text are moved to new places:

Text in lines 114-128 is rewritten and transferred to section in lines 113-125.

Text in lines 217-221 is rewritten and transferred to section in lines 183-186. 

Text in lines 222-224 is rewritten and transferred to section in lines 205-208.

  1. Line 65: Would you like to describe the shortest wall rule?

Response: I have described the shortest wall ruleb in lines 61-62

  1. Please explain the zippering, catastrophy and crossover of microtubules.

Response: I have described the zippering, catastrophe and crossover of microtubules in lines 114-117.

  1. Would you like to explain edge-catastrophe and enhanced stability?

Response:  I have described the edge-catastrophe in lines 120-122, however I think that term “enhanced stability” is understandable and does not require further explanation.

  1. Line 112: Which other mechanisms do you propose?”.

Response: I meant zippering, microtubule catastrophe and crossover. To describe it more clearly I have rewritten the text in lines 114-128 and moved to section in lines 113-125.

  1. Line 151: it is not clear. With the same orientation?

Response: I am sorry but your statement is not clear for me.

  1. Line 198: In which way, the reduction of katanin activity would be responsible for the shift in the orientation of what?

Response: The statement in this line was improper. I have rewritten this sentence (please see lines 195-197).

  1. In Fig. 1-B. It is difficult to understand the role of katatin. Is it activated at low Ca2+ concentration?

Response: yes, katanin activity is maintained at low concentration of calcium ions. I explained it in figure caption (line 214-216)

  1. line 201 and 202.This idea is not clear.

Response: I have rewritten sentences in lines 195-202. I hope that the idea is clear now.

  1. line 392: Check the English.

Response: I corrected the typo (now line 394)

Reviewer 2 Report

In this article, the author presented a review reporting the important role of auxin on cell division during plant enbryogenesis and discussing the possible mechanism of auxin-dependent cytoskeleton. The work was well organized and scientifically sound. My comments are as follows:

  1. Avoid using “I propose that ”, “I hypothesize that...”or “I propose a model...”, these sentences appeared many times in the article.
  2. Line 159, “To my best knowledge, there are no studies...”, please avoid using “to my best knowledge”, change it with other words to describe your content.
  3. Line 261, “I propose a model in which there are two stages of calcium accumulation in cytoplasm”, are there any experimental data to support your hypothesis?
  4. The change of calcium content may lead to calcium signal transduction which can be sensed by calcium sensors such as CaMs, CBLs and CDPKs. How about the roles of these calcium sensors in plant embryogenesis, please give examples.
  5. As for the model, the author should provide a model presenting the relationship between auxin and cytoskeleton during cell division.

Author Response

Dear Reviewer,

I thank for the critical reading of the manuscript and your comments which made me think about some issues. According to an Academic Editor some changes in the manuscript have been made. Please be also aware that in Figure 3 I changed depiction one of the symbol from PIN7 to PIN.

Furthermore, all changed and rewritten parts of the manuscript are highlighted. Some blocks of text are moved to new place:

Text in lines 114-128 is rewritten and transferred to section in lines 113-125.

Text in lines 217-221 is rewritten and transferred to section in lines 183-186. 

Text in lines 222-224 is rewritten and transferred to section in lines 205-208. 

  1. Avoid using “I propose that ”, “I hypothesize that...”or “I propose a model...”, these sentences appeared many times in the article.

Response: I agree that too many hypothesis have been made which may cause a lot of confusion and give impression of a very speculative content. I have rewritten some parts of manuscript to avoid using “I propose”, “I hypothesize”, etc.

Furthermore, I have cited more articles which support the presented content. Please see following citations:

-line 192 – citation [75]

-line 240 – citation [90,91]

-line 252 – citation [100] and [101, 102]

-Line 363 – citation [129]

  1. Line 159, “To my best knowledge, there are no studies...”, please avoid using “to my best knowledge”, change it with other words to describe your content

Response: I have rewritten the sentence in line 159, now line 157

  1. Line 261, “I propose a model in which there are two stages of calcium accumulation in cytoplasm”, are there any experimental data to support your hypothesis?

Response: According to suggestion of the Academic Editor I have dropped this hypothesis and rewritten this part of the manuscript (now line 254)

  1. The change of calcium content may lead to calcium signal transduction which can be sensed by calcium sensors such as CaMs, CBLs and CDPKs. How about the roles of these calcium sensors in plant embryogenesis, please give examples.

Response:  I have touched this issue a little. Calmodulin and CDPKs are mentioned in lines 252, 357, 372, 380

  1. As for the model, the author should provide a model presenting the relationship between auxin and cytoskeleton during cell division.

Response: I decided not to copy the model of the relation between auxin and cytoskeleton since it can be found in other papers. Figure 1 presents the relationship between calcium and cytoskeleton and in the text it is described that auxin induces calcium ions (which is presented in Figure 2).